# An Investigation into the Perspectives and Experiences of Physically Active Adults During the COVID-19 Pandemic

**DOI:** 10.3390/bs15050598

**Published:** 2025-04-29

**Authors:** Andy Pringle, Evelyn Oldale, Ella Rowley, Clare M. P. Roscoe

**Affiliations:** Clinical Exercise and Rehabilitation Research Centre, School of Sport and Exercise Science, University of Derby, Kedleston Road, Derby DE22 1GB, UK; evelynoldale@gmail.com (E.O.); c.roscoe@derby.ac.uk (C.M.P.R.)

**Keywords:** physical activity, actives, barriers, facilitators, COVID-19, CMO guidelines, intervention, messaging, guidelines, mental health, qualitative research, implementation

## Abstract

Contemporary physical activity (PA) strategies emphasise the PA needs of sedentary and inactive groups, with less emphasis placed on physically active groups. Understanding the needs of physically active groups is important in helping people to keep active. This study investigated the perspectives and experiences of physically active adults (‘Actives’) during the COVID-19 pandemic, including their PA levels, barriers and facilitators to/for PA, the strategies they deployed to keep active and their experiences of the messaging of Government health and PA guidelines. Following recruitment, thirteen in depth semi-structured interviews were undertaken with adult men and women who reported meeting the UK Chief Medical Officer’s PA guidelines before the COVID-19 pandemic commenced. Braun and Clarke’s thematic analysis identified five key themes and related sub-themes: (I) PA participation; (II) barriers to PA participation, including overcrowding of the PA space, conflict between different groups and negative mental health; (III) facilitators for PA, including place/residence, rural location, social support and good mental wellbeing; (IV) strategies to keep active, including improvisation, substitution of PA mode, scheduling PA, social support and goal setting; (V) guidance and messaging on the health guidelines, including PA promotion for strength and balance, mental health and where to receive information on PA. This study provides valuable insights for PA promotion for Actives at an unprecedented time.

## 1. Introduction

The [46] ([46]) recommended that adults 19–64 years of age should undertake 150 min of moderate physical activity (MPA) or 75 min of vigorous physical activity (VPA), or a combination of the two, each week, as well as undertaking physical activities that maintain strength and balance 2–3 times per week. The beneficial role that physical activity (PA) can have for the prevention and management of long-term conditions, including cardiovascular disease (CVD), mental health and some cancers has been reported ([46]). Before the COVID-19 pandemic, a third of adults reported not meeting the recommended guidelines ([40]). Physical inactivity is also prevalent among UK older adults, with 27% of 55–74-year-olds and 47% of 75–84-year-olds completing less than 30 min of PA weekly at the end of 2020 ([44]).

Although it is important for people not meeting the guidelines to be physically active ([46]), it is also important to support those people who are currently physically active, including those meeting the guidelines to maintain their participation. This is in order that they continue to obtain the well reported health and social benefits. Helping people to keep physically active and reducing the risk of health conditions ([46]) also provides potential cost savings to public health services. Given the health benefits of regular PA participation, and that insufficient numbers of people are meeting the PA guidelines ([46]), past and contemporary PA strategies have and continue to focus on the needs of sedentary and inactive populations, helping them be active ([33]; [43]; [46]). This includes the period following the COVID-19 pandemic ([43]). However, in these strategies and interventions, comparatively less emphasis is placed on the needs of physically active groups and what helps and hinders them to maintain the recommended PA levels, including following the COVID-19 pandemic.

The year 2020 was characterised by the emergence of the pandemic caused by the spread of the COVID-19 virus, followed by public health interventions and measures to manage the effects of the virus globally, including in the UK ([35]; [49]). On 23 March 2020, restrictions were imposed by the UK Government to prevent the spread of the COVID-19 virus. The government restrictions led to new laws impacting how people lived their lives, instructing people to stay at home for all but essential reasons, including essential work; caring for animals, including walking a dog; food shopping; caring for vulnerable people; accessing health and social care and undertaking PA ([13]). To enforce the restrictions, police forces were issued with new powers to ensure social distancing using on-the-spot fines, and to direct or remove people to their homes, by force if necessary ([9]). The restrictions and ‘lockdown’ led to disruptions to people’s lifestyle habits, including their PA patterns ([32]; [35]).

Under the UK Government guidelines, this meant that people were normally permitted to leave their home once per day to undertake PA, although this could be more frequent if people performed PA as part of undertaking essential activities such as commuting to essential work or caring for family members or friends ([35]). At this time, restrictions extended to the access of PA opportunities, with non-essential services such as gyms, sports centres’, recreation and health clubs ([35]), some parks and green spaces ([12]) being forced to close their doors/gates during the periods of government restrictions. The restrictions in the leisure sector led to a temporary cessation of ‘face-to-face’ sport and recreation services, meaning that for many people who use these services, their PA routines and networks which support their health could not be undertaken as per usual, thus meaning that PA routines were negatively impacted ([48]). Further, the closure of some green spaces such as parks and the grounds of stately homes and historic monuments ([12]) also reduced opportunities for PA. The closure of non-essential retail, hospitality and entertainment sectors led to restrictions on people’s leisure activities, including bars, cafes, restaurants, arts centres, cinemas, theatres, attending live sporting events and shopping. However, some retail shopping classed as ‘essential’ remained open, including food shopping; car maintenance; DIY stores; sport and exercise retail, extending to fitness equipment; bicycles; clothing and footwear. These items were made available through online providers, as well as some essential shops and services that remained open with the necessary safety restrictions associated with delivery and collection of goods. During the pandemic, there was a rise in the demand for sports equipment, such as bicycles as cycling increased ([8]); fitness equipment, including home gym equipment, weights and resistance bands; as well as clothing and footwear. Further, manufacturers and suppliers had trouble obtaining raw materials and finished stock due to decreased production, increased demand and reductions in their supply chains ([42]).

With the closure of PA services and sport and leisure facilities, physically active people sometimes substituted their ‘usual mode and place’ of PA which was not available with a replacement activity ([5]). For example, a person who would normally go to the gym might replace this with going for a run or performing some circuit training at their home using equipment they had purchased or already owned. There was also an increase in the numbers of sedentary and inactive people undertaking PA during this period, in some cases, substituting their sedentary leisure pursuits that were not available with PA participation. Research identified that despite the closure of sports facilities and restrictions on non-essential travel, the COVID-19 lockdown led to increases in interest in PA ([8]) and engagement with PA ([15]).

During this period and in the UK, it was not uncommon to hear reports of overcrowding of the PA infrastructure. Spaces and places commonly used for PA include green spaces ([37]), e.g., urban and forest parks, leisure farms ([50]), a green network of walking and cycling routes ([12]), river and canal towpaths, flood plains and footpaths. In some cases, blue spaces such as rivers, waterfronts, beaches and lakes ([18]), and some grey spaces such as car parks, games areas and other municipal spaces were not available due to the restrictions. The oversubscription of PA spaces was exacerbated by government restrictions, such as the closure of sport and leisure facilities and guidelines instructing people to ‘stay at home for all but essential reasons’ combined with the directive to ‘stay local’ to their area, and to only ‘leave the house once per day’ for PA ([13]). This resulted in high numbers of people being concentrated in smaller local geographical locations for the purpose of undertaking daily PA, leading to overcrowding ([19]). [19] ([19]) highlights that a common barrier was a lack of space and appropriate facilities, making it more difficult to stay active during the closure of gyms and many parks and trails. While some people have access to gardens and yards, there were others who did not, placing greater importance on the availability of local spaces for PA. Further, vulnerable people were less able to use parks and green space in the lockdown, having been told to self-isolate as part of government restrictions ([12]).

[48] ([48]) reports that the pandemic highlighted how residents in inner cities do not always have green spaces nearby. [12] ([12]) also inform us that not having green space within walking distance is acutely felt as people have been unable to travel as a consequence of the COVID-19 pandemic; therefore, they were limited to the facilities available locally. In contrast, people living in remote and rural areas reported that the restrictions were not as impactful because they lived in sparsely populated areas and had access to green space. These areas were not normally accessible to people living outside of the locality because of the government restrictions on travel, and they were less likely to be subject to monitoring by statutory bodies charged with enforcing the guidelines ([13]).

The concentration of a greater number of people into fewer local areas led to reports of overcrowding of the PA infrastructure ([41]). It was not uncommon to hear personal reports of frustration and confrontation between user groups because of a lack of protocol and etiquette. For example, this included confrontation between different PA groups, e.g., walkers, cyclists and dog walkers, as well as conflict between novices and experienced exercisers. Sources of confrontation might include people walking or cycling side by side and not sharing the space with other users; people loitering; unintentionally blocking infrastructure such as pavements, resulting in congestion; cyclists riding too fast in shared paths impacting other active users; people wearing headphones and not being aware of what was going on around them, including their proximity to other users.

Although there were challenges associated with keeping physically active at this time, there were opportunities for PA ([35]), and the outbreak of the pandemic accelerated the use of modern technology in the fitness sector, including live exercise classes delivered by a range of providers through online platforms ([30]; [35]), such as Zoom and MS Teams. This provision also included live virtual events, such as Joe Wicks’s daily fitness workouts broadcasted via his YouTube channel for five days per week for eighteen weeks, which was aimed at keeping young people active ([6]) as well as commercial online offerings such as [31] ([31]) and [51] ([51]). Zwift is an ‘online exercise software application that enables an individual to cycle, as a personalised digital avatar, with others in real-time’ ([38]). Other online PA options included exercise providers offering their usual exercise classes virtually and via pre-recorded PA sessions that could be accessed via online platforms at a time of convenience to the user, such as on YouTube ([35]). Further, illustrations of exercises/exercise movements that people could download, were also made available by charities, allowing people turning to these platforms to keep active during this period ([35]).

During COVID-19, the focus was on helping sedentary and inactive people to be active during the pandemic ([45]); however, it is important to help active people to continue to meet the guidelines. This is not only important from a public health perspective, but also from an economic perspective, given potential savings to health services ([33]). It is important to consider the barriers and facilitators that active people faced keeping active during the COVID-19 pandemic and periods of restrictions and the strategies they used to keep active and their perspectives on the messaging of government health and PA guidelines. This can provide useful information for shaping interventions that best meet their needs and help them to remain physically active during such times. With this in mind, this study investigated the perspectives and experiences of physically active adults (‘Actives’) during the COVID-19 pandemic, including their PA participation, barriers and facilitators to/for PA, the strategies they deployed to keep physically active and their experiences of the messaging of Government health and PA guidelines.

## 2. Materials and Methods

### 2.1. Recruitment and Sample

In this study, participants received pre-information on the purpose of the study and their rights and freedoms, including their right to withdraw. All participants consented to take part in the study. To meet the aims and objectives for this study, we deployed an in-depth semi-structured interview, with a convenience sample of 13 adult participants aged 18 years and older. Participants were invited to take part in the study if they regarded themselves as being physically ‘active’ (i.e., reporting meeting the UK CMO PA guidelines, 2019, of 150 MPA/75 min VPA or a combination of the two per-week ([46]) and had been doing so for the period from 1 October 2019 to 20 March 2020. The invitation to take part in the study was advertised through word of mouth and promotions through PA networks and social media using promotional materials. Following pre-information and consent, participants were invited to take part in a semi-structured interview led by members of the research team. Interviews were undertaken on MS Teams and recorded. This approach has previously been used to effectively collect information on the PA behaviours and the barriers and facilitators of Actives, including during the COVID-19 pandemic ([35]; [39]). A copy of the interview schedule is available in Appendix A. If participants preferred, a telephone interview or in person interview was offered, and this was recorded using a digital voice recorder. Notes were taken if participants did not wish to have their interview recorded. Participants received no remuneration to take part.

### 2.2. Instrumentation and Data Analysis

A qualitative interview approach was adopted in this study, and this was used to obtain rich informative accounts of the PA characteristics of participants, including their experiences, barriers and facilitators to PA. We adapted an interview schedule from already published sources in the peer-reviewed literature ([35]). Information was collected on participant demographics (e.g., age, gender, ethnicity); a verbal account of participants’ PA engagement with the UK CMO Physical Activity Guidelines, 2019; barriers to PA participation; and facilitators to PA participation and government guidelines. We also collected information on the strategies that active participants used to keep active during the COVID-19 pandemic and associated restrictions, and their experiences of the governmental health and PA guidelines, including messaging. Prior to data collection, the instrumentation was piloted, and the research team practiced their interview technique, ensuring that they had refined their interview technique and could facilitate effective data collection. Inductive categorization was carried out in this study and data analysis was undertaken using [7]’s ([7]) six stages of thematic analysis. That is, following transcription, interviews were read through to saturation and then interesting features of the data were identified into codes. Following this, codes were grouped into themes/sub-themes, and these were then used as an organizing framework to report the research findings in consideration with the objectives for this study. To confirm the themes, three researchers (AP; EO; ER) met to discuss and review these for consistency and representativeness. This approach has been undertaken by the author in previous research studies ([35]; [39]). Table 1 provides an overview of the themes and alignment with the subthemes.

## 3. Results

Participants were white British (n = 13), women (n = 7) and aged 18–24 years (n = 2); 35–44 years (n = 2); 55–64 years (n = 6); and 65–74 years, (n = 3). The age categories ranged from 18 to 24 years through to 65 to 74 years. Following thematic analysis of the interview data, five key themes along with related sub-themes were confirmed (Table 1). A selection of the excerpts is provided to support the themes in this results section. The average interview time was 24.49 min.

### 3.1. Physical Activity

#### 3.1.1. Physical Activity Levels

Actives reported their PA participation; many participants reported running, cycling, walking and orienteering. Some participants reported they undertook PA that required facilities, such as swimming, exercise classes and gym sessions. Only occasionally did participants report participating in team sports, such as netball or specific activities that developed strength or activities, classed as HIIT.
*I would cycle four to five times a week for about one and half to two hours, longer at the weekend, and walk in between*.P7.
*I would be doing short interval training, hill sprints and smaller, quicker runs like 5Ks and then usually some sort of form of longer run. That could be anything sort of 10 K to half-marathon distance and in between that I would be cycling for commuting purposes and to work 15 miles and with 30-mile round journey*.P8.
*I cycle to work when I’m in the office, so that’s about 15 miles each way. I swim probably three times a week. On the morning, I go to the gym two or three times a week before swimming as well. I do a yoga class one evening. I also do a couple of HIIT classes. I try and run twice a week, a long run normally on a weekend for several hours*.P5.
Participants reported how COVID-19 and the restrictions impacted their PA participation, and this varied between respondents.

#### 3.1.2. Participants Reduced Their PA

*Because of the restrictions, I had to modify my activity; before I would go out on my bike, then go for a walk later, I could not do that. Instead of going on the bike for 2–3 h, I went for an hour because that is what the restrictions said. So actually, my PA will have gone down because of this*.P3.

#### 3.1.3. Participants Substituted Usual PA with Another Mode

*I stopped cycling and it was really funny because we live in a little village and actually having no traffic. So the very strict period of lockdown was actually lovely because people weren’t that fussed about whether you were outdoors or not. You know, we weren’t sort of spying on each other. There was lots of space, there was no traffic. So you could take the dog for a walk without being run over. And it was really quite nice. So I just did that rather than cycling*.P2.

*In the three months before the pandemic, there was a lot going on. I have to be honest that actually I lost motivation and apart from the dog walking because I had to do that, I actually didn’t really do anything at all. So probably, the pandemic provided an opportunity for me to actually get back to doing a lot of the things [PA] that I actually used to do, or some of them, obviously because there was some restrictions*.P6.

*I think there were a lot of positives and I think for me that kind of set out a whole new routine about how I am now physically active compared to the periods of leading into the pandemic. I was working from home, so needed to break up sedentary behaviour as well as undertaking PA*.P7.

#### 3.1.4. Participants Increased Their Physical Activity

*If anything, I would say perhaps I did more activity because I had more time at home, just not necessarily the range of activities that I was doing before when everything was open*.P5.

*My daughters were back here [during the first lockdown], we were going on quite long family walks and one of the daughters wanted me to run with her. And she’s sort of a little slower than me at the time. So, you know, I sort of felt the need to go running on my own as well. So yeah, I was doing a lot of activity. Then, when things eased, it decreased because I went to look after my dad for a bit and I volunteered with asylum seekers in Derby and that resumed*.P2.

*We did the sections of the Midshires way from home and using public transport because obviously it goes from sort of Stockport or somewhere right down to [Buckinghamshire], we just did the Derbyshire sections, and we got all the way south of Belper to the border of Derbyshire and Leicestershire. So we must have done about 50 miles of that in sections because three of us would run it and then we’d have to either run back to where we left the car, or run to a different bus stop or something, you know, so that was more [PA]*.P1.

### 3.2. Barriers to Physical Activity Participation

Participants reported some of the barriers they encountered in maintaining their PA participation.

#### 3.2.1. Closure of Spaces and Places for Physical Activity

*So during lockdown, the local leisure centre, where I go for swimming and for the fitness classes, wasn’t open. However, I would say that the main impact was on those activities which are more further afield. So for example, we couldn’t go away on skiing holidays. We couldn’t necessarily go up to Scotland sea kayaking*.P5.

*The only thing that would impact on me then [was] just my own time constraints. Working from home—moving more into a virtual space…. so, suddenly, your days are back-to-back. The main change for me was just the commuting aspect on the bike really because of the rules. Working from home/establishing a routine that favoured work and not PA*.P8.

#### 3.2.2. The Lack of Availability of Products to Support PA

Participants also expressed some challenges with accessing products for PA.

*You could not buy a bike I could ride on the road; you could not buy a road bike for love nor money, the shops were closed, Halfords had click and collect, it was absolutely sold out of everything, equipment, components, it looked like it was closing. It was frustrating that it held me back*.P3.

*Some of the instructors did not want to run the exercise sessions face to face at the leisure Centre, but they were running them online, I tried to get hold of weights and then, you know, kettlebells and things like that. And it’s just impossible because everybody had obviously gone online and ordered them. Yeah. And everyone everywhere was out of stock*.P5.

*I remember going to a bicycle shop and basically the guy in the shop said, ‘every day is like a Saturday, every day is like Christmas Day, we have such a demand for bicycles.’ You could not get stuff! People like to spend, they like to consume, they like to shop, and the process of shopping, I think it’s kind of it’s one of the nation’s leisure activities it’s one of the things that people do they like to consume, they like to shop [and] the shopping experience, and so I think people who typically wouldn’t be going out to buy a bike or exercise equipment to get kind of walking boots and walking trousers, they like that experience of going to shop more than they do with the reality of going out to exercise*.P7.

#### 3.2.3. An Overcrowded Physical Activity Space

The concentration of greater numbers of people into fewer spaces also created several challenges and barriers to participation in PA.

*We did take the dog down to the park and it would be absolutely rammed with people, and you could tell the regulars were not happy. The regular dog walkers are very vocal about people using ‘their space’. For me it was great to see people being active and people out walking. You know, particularly the walking stuff and I thought I might see more people out walking now. I don’t think I’ve seen that before*.P6.

*I was running with my friend, and this woman had her dog on an extendable lead that got tangled up in her legs, my friend said, ‘get your f***ing dog under control’, the women said, ‘don’t you speak to my dog like that’*.P5.

*If you did go on a walk, the routes were absolutely rammed, and you couldn’t move for people. I didn’t always feel necessarily safe from the kind of COVID-19 space*.P10.

#### 3.2.4. Novice Exercisers and Not Sharing Space

Tensions arose between novice and regular exercisers and the feeling people were not sharing the PA space.

*There were a lot of problems around here with the cyclists, you do get a lot of more serious cyclists around this area doing the Tour de France routes, you know, the Grand Departee routes, they [the experienced cyclists] were very frustrated with some of the beginners that were out there having a crack and I’m sure it wound them up*.P6.

*You had a lot of newbie cyclists around here, you cycle typically on the left and so they would cycle side by side, they wouldn’t give way and share the space and so there is all sorts of challenges around that etiquette, so that kind of community of sharing the infrastructure and that kind of rights, perceived kind of entitlements, that space was theirs so they felt [they] didn’t have to share it*.P7.

*The amount of new people that were out on the trail, not being able to social distance, worrying about the virus and so I could keep myself safe and keep my relatives safe. I was frustrated by people who were using the trails and not sharing with care for others and not keeping to the left, I wanted to paint the trails to tell people to keep to the left, so everyone could use the paths. There were lots of reasons to stop you being active, because of other people’s behaviour*.P3.

*Some of the nature of the activity that I always [do] independently, if that’s out on the bike or out running, then you’re not really in a fixed place. So even if it was busy, you’re passing through it. So, if I’m running on the road or a path or through a trail, you’re not there for any great length of time. So, it was never really an issue for me*.P8.

*I remember cycling on what is a cycle path but it shared with pedestrians and this woman telling me off for cycling on the pavement and I thought, well you quite clearly have not used this infrastructure previously, because if you did you would know that this is actually a cycle lane and you [are] probably kind of using this for the first time and not [realised] that it’s shared infrastructure*.P7.

#### 3.2.5. An Unwelcoming Space for PA

In some cases, our participants encountered difficulties with residents who felt they should not be using the PA space.
*A few times where by you be made to feel unwelcome in certain areas I got the feeling that you know you were trespassing; even though it’s like a public bridleway, you could see the kind of looks of disapproval that you were kind of out walking at the same time as them, or doing activity at the same time as this was their time to exercise*.P10.
*When I was walking on the street, I mean, I walked down the street one day, and a couple came the other way and the man pushed his wife or partner into the road to avoid me*.P9.
*I was cycling and there were three men cycling side-by-side, I rang my bell, as to get past and they refused to move and when I passed, they called me a f***ing bitch*.P3.
*I’d come by here [the cycle path on the canal] it was closed by the Canals and River Trust, someone put a big barrier up. The canal boat owners would look at you, like you should not be here. They would jump out and say you ‘should not be here’. I thought, you can go and **** yourself*.P13.
In some cases, participants reported that some local people were policing the infrastructure.

*Local residents had actually got hold of some of the police leaflets…. which were saying ‘you know you shouldn’t be here’, ‘the guidance means that you shouldn’t be in this location’, ‘the rules are very clear’, they had copied and blown them up and laminated them and put them up, what I would consider to be illegally on the side of the road*.P7.

*I had no encounters [with people], but there was a lot of signs put up that strictly weren’t legal. You know, there’s a lot of public footpaths near us that go kind of on lanes, past houses and I mean, obviously, you know, you’re not going to cough on people’s doorstep. You just gonna pass by as quickly as you can. So that was a bit annoying and then farmers putting up signs by styles and things. You know, it’s just a bit. Everyone went a bit mad there, I thought*.P1.

*There were signs up, saying go home, we don’t want you here. Go home. And there’s one sign which actually said we will report you to the police—and I just thought, is this the Stasi operating here? This is unbelievable that people are enjoying trapping people who wanna just be out in the fresh air*.P9.

*I had a confrontation with a guy who said I shouldn’t be cycling there and that there was a sign he said had been put up by the police saying that you shouldn’t be here and so I think it then created a lot of confrontation of people in local areas and telling people that they should and shouldn’t be in certain locations, you could see them kind of making comments to the people they were speaking to, kind of suggesting that they should be there*.P7.

*I heard reports in the Yorkshire Dales, that locals were putting tacks and glass on the road so cyclists would get punctures, to try and deter them from going there*.P5.

#### 3.2.6. Loss of Social Support for PA

Participants reported how losing their social support was a barrier to participation during the first lockdown and following the easing of restrictions.

*I’ve got [a] friend I cycled with, sadly, he’s died from COVID-19, which is solely tragic, because he went in for a [minor] operation and [he got] COVID-19 and he never came home. So, I’m afraid that’s one of the statistics, so I lost my best mate, we cycled together*.P9.

*Through lockdown, I mean one of the first activities I was getting back into before the pandemic was going back to netball and I think I needed the social element of that. And so, I went once, and then we went into lockdown. So that was a big barrier for me. I lost out on the activity and the socialization process. My daughter went back to London after the first lockdown, and it was a little bit more difficult to get the motivation to be active*.P6.

### 3.3. Facilitators to Physical Activity Participation

#### 3.3.1. Making the Most of the Opportunity to Be Active

Participants reported ‘making the most of the unique opportunity they had to be active’.


*It [PA] was very precious, you literally had an hour to go out on the bike, it was very frustrating not being able to go for a walk later, so it [PA] was very precious!*
P3.

*I was conscious of how fortunate we were in living where we are, in being able to continue the activities that we were used to doing*.P5.

*COVID-19—it’s made me more active today because it’s just made me, I guess, admire it [PA] more and like just doing PA*.P4.

#### 3.3.2. Place of Residence

Several participants lived in locations which were rural or where they had access to spaces for PA, including those that were isolated and or/not being policed by the authorities.

*Well, I just ignored that one hour a day rule. Cause who’s gonna check on you? You know, I live sort of on the edge of Belper. And honestly, no one is gonna check on you*.P1.

*I literally live just on the edge of the Moor and so it didn’t actually alter how many times I went out and I didn’t really give it any thought, to be honest. And if I’m honest, it didn’t actually stop me. I went out as often as I wanted to*.P5.

*I was party to lots of discussions about what people should and shouldn’t do and I was very well aware of the issues about driving to places and whether that was OK. I personally wasn’t tempted to do that because I got a good variety of what I could do where I live, and because I was challenging myself to build my distances, I did get up to 15 K, which meant that I was running*.P11.

*Yeah, so I come from a tiny village near Scarborough. So there’s loads of back roads and stuff for cycling*.P12.

#### 3.3.3. Additional Permissions to Leave Home for PA

In some instances, participants reported that they had additional permission to leave the home, and this was an opportunity for PA.

*I was lucky I didn’t just adhere to the one hour a day because I had a dog. I was allowed to do more*.P6.

*We kept taking the ponies out, so I changed what we do in the ponies cause I would normally lead my pony, and my mum would ride and instead of taking my pony and hand, I just walked with my mum so she could keep riding. I realised I was very, very fortunate having the animals in the space and stuff. I’ve got a colleague in a flat in London who, you know, had none of that. The animals, by their nature, force you to get out, you know, to do stuff*.P2.

*I made sure I prioritise my PA so I went out and exercise[d] once a day, but I also used some of the local services for like food, so I would actually go and walk down to the butchers for 6:00 o’clock and I would get my shopping before it got busy because it’s only a small shop and I was worried about contracting COVID-19 and then I would actually take a more circuitous route back home to legally permissively do PA, but just do a longer route*.P7.

#### 3.3.4. Mental Health

Participants also reported that good mental health was a reason to be active

*Probably the only other barrier for me at that point was my own mental health, and so my head was all over the place and that was a massive barrier. When the sun [was] shining and you know it’s a great place to be and in the middle of nowhere, but for me as well and my mental health getting outside is really important*.P6.

*I don’t know if this adds up, but when we were in lockdown, it was like we were in an exciting period and actually it was great to go out and do something. Here am I with nature with birds, with the wildlife and the countryside and so mentally, I kept fresh*.P9.

#### 3.3.5. Weather

Participants reported the importance of the weather as a facilitator.

*The weather was good and nice and quiet, so no fears of getting COVID-19*.P6.

#### 3.3.6. Social Support

*We came out of the first lockdown in the start to come out of it in the June of 2020, and we had some increased freedom. So when you had increased freedoms, you could go and visit people in the first of all and sort of smaller groups. I then went back to netball and that was really good and really important to me to get back into that*.P6.

### 3.4. Strategies That Actives Adopted to Support PA Particpation

Participants reported on the strategies they adopted to keep physically active.

#### 3.4.1. Engaging in Physical Activity Challenges

*I was using the bleep test as a way of just doing a bit of exercise and then maybe just having one or two beers afterwards…. just trying to get some of that social aspect into it*.P8.

*There’s a company called Race, that you sign up and you say I’m gonna do X number of kilometres or miles and then they give you a medal at the end of it*.P2.

#### 3.4.2. Setting Goals for Physical Activity Participation

*In forced liberation from all sorts of competing activities, I aimed to build my running so I could do 17 K again by my 70th birthday, I didn’t quite achieve it, but basically, I treated that first lockdown as an opportunity to actually stick carefully to my goals*.P11.

*I lost my husband at the start of the year [2020], so we set ourselves some family challenges with the goal of raising money for his charity, which made me go out and do stuff*.P6.

#### 3.4.3. Keeping Physical Activity Interesting

*I ran and often took orienteering maps with me to look out on the way, just to keep your brain ticking over with the navigation and things*.P1.

#### 3.4.4. Online Physical Activity Provision

*I did the NHS fitness videos, so I rolled up the carpet in the lounge and did that, even though I am not an indoor exerciser, just so I was doing some exercise*.P3.

*I purchased a turbo train[er] online, so when I got home from holiday it was there and then I just got into the habit of maybe doing less running and going on the turbo trainer and doing a GCN [Global Cycle Network] class*.P8.

*I couldn’t actually go to the leisure centre to swim or do the fitness classes, so I did the classes online at home*.P5.

*In the context of lockdown, the zoom classes were fun. You know you logged on 5 min early and waved at your friends and what have you. So in terms of my activity, it became something I could focus on even more*.P11.

#### 3.4.5. Skills and Processes to Support PA

##### Improvisation/Contingency Planning

Participants improvised to keep active:

*I cleared the path round the whole of the perimeter of my garden, which is on a steep hill, and I was still able to be active*.P11.

*One of the skills that you have as an exerciser are…thinking about how you contingency plan how you organise things so you’re able to prioritise your physical activity. I have a series of blocks where I live, so I walked around small and bigger blocks to break up my sedentary time*.P7.

*I would run round my golf course, it was closed with a couple of hills in it, and I loved every minute of it because I was free. It was quiet. I felt free and open. It was fantastic*.P9.

Scheduling physical activity into a daily routine:

*I would go out cycling earlier in the day, when it was quieter and safer and so had [to] modify things that way. I would exercise in a way that would keep me fit, instead of having to slow down for every person on the trail*.P3.

*I was getting up at 5 am going for a run, coming home and starting the school day with my children and then doing Joe Wicks*.P8.

Accountability to exercise with others:

*My daughter moved into my home during the first lockdown and she hadn’t really been a runner but started running as well. So, there was that accountability with her in there. Was somebody saying no, you can’t have your tea until you’ve been out and done whatever you’ve done. Having a buddy or someone who checks in on you and pushes you and makes you accountable. I think it’s really useful*.P6.

Self-monitoring techniques:

*I was monitoring my activity levels [participant shows me their logbook detailing their activity], it certainly helped me. It certainly motivated me to go again. I’ve got something that I can show anybody that, you know, prove I didn’t do bloody nothing*.P9.

Selecting and finding new locations to exercise:

*When we went for walks, we didn’t go to sort of real honey pot areas in Derbyshire, you know, we just stayed in like the local woods or whatever, that was one thing that we did [which] was really good*.P1.

*I lived in close proximity to open spaces, and I would go a bit more off the beaten track and kind of find things that nobody else would be doing. I was looking for new places to go to look for less crowded places and had a much better appreciation for the countryside and the local area*.P10.

#### 3.4.6. Keeping a Low Profile to Avoid Conflict with Local People

In reducing the risk of confrontation, some participants reported keeping a low profile.
*At the Monsal trail, we would have our sandwiches. Um, you know, carefully out of the way. We didn’t sort of sit there and make it as though we were wanting attention from locals*.P9.
This contrasted with some reports of other people’s actions:

*People had driven to the coast or driven to local beauty spots and that could not necessarily be kind of classed as being local, there was an element of inequity and injustice. There was a bloke who drove up from the southeast to the Dales with a kayak on the roof and the police turned him back*.P7.

### 3.5. Government Guidelines and Messaging

#### 3.5.1. The Guidance Could Be More Helpful for Promoting PA and Health

*I don’t think they [Government] were explicit enough about, you know, what are those [PA health] benefits and I don’t remember seeing very much about it. So very well saying this is good for you, that’s good for you, but actually some guidance on how to do it. You know, some directives to actually where you can actually access these particular things. I think there could have been some kind of resource to actually help people with that as well*.P6.

*All those people that played team sports… who really aren’t into running, then? Their PA levels potentially just drops, and I suppose it’s trying to think about what the messaging was for that group in the middle as opposed to the inactive and active one*.P8.

#### 3.5.2. A Lack of Guidance on Strength and Balance Training

*They missed that message around the strength and balance stuff that actually needed to continue. I don’t think I heard anything about that message, for older people in particular*.P6.

#### 3.5.3. A Lack of Guidance on the Mental Health Benefits

*I don’t remember seeing anything about the particular mental health benefits, the emotional wellbeing of activity*.P6.

*Being told how many times you’re allowed to go outside today, and how that impacts your kind of mental health, your psychological state, being able to set goals based on that kind of stuff*.P10.

#### 3.5.4. A Lack of Clarity on What Was Defined as ‘Local’ Exercise

*I think there was a real lack of detail in terms of what you could do, or what you could do in terms of the notion of what local was, how local was defined I think this led to a lot of ambiguity. I think it also led to a lot of anger*.P7.

*I think for people who were typically inactive, I think more guidance for them would have been beneficial, at the same time that is kind of nagging feeling in the back of [your] head that probably [you] shouldn’t be doing this*.P10.

#### 3.5.5. Guidance on How Often and Long People Were Allowed to Exercise Was Interpreted Differently

Some people reported that the guidance on the restrictions was not relevant.
*I didn’t change anything, I rode my bike for 15 h a week. I just did the same as what the training plan said, the guidelines didn’t bother me cause I still done the same thing anyway*.P12.
*I just ignored that one hour a day rule. Cause who’s gonna check on you*?P1.
*I went out on my bike, for two hours a day, every day*.P13.
Other participants adhered rigidly to the guidelines of exercising once per day.

*I would go out on my bike for one hour, because that is what the restrictions dictated to us*.P3.

*There was a lack of consideration and a lack of etiquette on sharing the space in some cases you had people who were going out and being physically active in groups when they weren’t legally permitted. I think if you can’t adhere to the guidance, that became a source of frustration. Lots of people have not met their friends and families and that was kind of a source of injustice*.P7.

## 4. Discussion

Our research set out to investigate the perceptions and experiences of Actives during the COVID-19 pandemic. The key findings from this study identify that: (I) Actives who reported meeting the PA guidelines made changes in their PA levels throughout the COVID-19 pandemic, with some people reporting both increases and decreases. The PA levels of participants also changed as the government restrictions were eased. (II) Actives reported encountering barriers to PA, including the COVID-19 restrictions, the overcrowding of the PA infrastructure and loss of social support. From the tone of some of the interviews, conflict and confrontation were one of the most intensely felt barriers for some participants. Actives offered accounts of conflict with other people who were being active, including a perceived lack of etiquette from novice/other exercisers. They also experienced disapproval and/or intervention from other members of the public who felt they (the Actives), should not be using the PA space, and in some instances, members of the public were viewed quite strongly as ‘policing the PA space’ and attempting to deter its use. (III) Actives reported facilitators to PA participation, including their place of residence, living in rural and remote areas, undertaking legally permissive roles, allowed under the restrictions which supported PA, such as dog walking, food shopping and caring for animals, as well as exercise. They also reported the importance of PA for mental health (IV) The strategies adopted to keep physically active included scheduling PA into daily routines, substitution of PA modes, contingency planning, improvisation and self-monitoring. (V) Government guidelines were not always seen as helpful when promoting PA, as they did not clarify the parameters of the restrictions, such as ‘to stay local’ or maximise the opportunity for PA promotion, including messaging on strength and balance, PA for older people, mental health and wellbeing and where people could find out how to be physically active.

### 4.1. Physical Activity Levels

Research has shown how the government’s COVID-19 restrictions have had a detrimental impact on people’s daily routines, including accessing health services ([20]), while research has indicated that the COVID-19 pandemic has resulted in reductions in PA levels ([40]). To participate in this study, participants were required to confirm in the interview that they were meeting the UK CMO PA guidelines of 2019, of 150 MPA or 75 min VPA per week or a combination of the two before the pandemic. Many of our participants reported how they undertook walking, running, aerobics, swimming, cycling or a combination of these activities. Fewer people reported participating in team sports such as netball, rugby or gym-based activities or HIIT. Participants reported making changes to their frequency, intensity, time and type of PA, including substituting modes or adding in different modes of PA due to the unavailability of their usual PA options. In some cases, the restrictions facilitated additional options for PA that were permitted under the restrictions such as food shopping and looking after animals which could be combined with PA. In this study, some participants reported that while their mode of PA changed, their PA participation levels stayed the same or even increased, with people maximizing the opportunities that presented to them due to their place of residence and or living in remote locations. Participants were acutely aware of the importance of maintaining their PA during the pandemic for their wellbeing, such as maintaining good mental wellbeing or reducing sedentary behaviours ([46]).

### 4.2. Barriers to Physical Activity

Research has identified that people faced obstacles in their ability to be active and reaching the recommended PA levels, and this may have cascaded to other dimensions of wellbeing ([26]). In this study, our Actives had a long-standing experience and history of being active and they had experience in refining the processes and strategies that support PA; this is something that we discuss in this section. In this respect, Actives also had the experience of performing their activities, they knew what it took to keep active and overcome barriers reported in the literature ([44]). That said, several of our participants reported barriers to PA participation. Transmission of the virus was a barrier to participation, although this did not stop Actives, who adopted strategies such as social distancing and waking up early or being active when spaces were quieter for exercise (e.g., cycling and walking in the early hours), and in some cases, actively travelling to and seeking out less-known places that were under populated.

Given the unprecedented nature of the pandemic, some Actives reported the loss of social connections that facilitated PA during the pandemic, including family members and their exercise buddies, who moved away; in some instances, participants had lost friends and their exercise partner due to the virus, while in other instances, family members had moved back home following the easing of restrictions, thus impacting their motivation to undertake PA. In doing so, participants had to cope with the loss of a resource and a facilitator that helped them maintain their PA. It is understandable, given the unprecedented nature of the pandemic, participants reported that at times, their mental health was a barrier to PA. Some participants reported not feeling this, but they all recognised the importance of keeping active for feeling better and improving their mental wellbeing, in some cases forcing themselves to be active. This is a resource that comes from the experience of being habitually active, that people can motivate themselves to undertake a behaviour, which will intrinsically make them feel better afterwards, even though they might not feel like performing PA at the outset.

Restrictions on leaving the home for essential work led to an increase in home working, and the change to working from home affected people differently, as they faced different home and work environments ([10]). The shift to working from home meant that some of our Actives lost the opportunity to perform PA through their active commute to work, volunteering or being active when at work. Work and transportation, along with leisure and domesticity, are domains that contribute to people’s activities ([3]). Research has identified that working from home resulted in loss of PA ([10]) and our Actives had to develop strategies for substituting their lost PA which they would ordinarily undertake travelling to and from work and while at work, another mode of PA. Several Actives reported difficulties with obtaining exercise equipment and spare parts due to the demand for such items and reductions in availability as reported earlier, necessitating improvisation and adaptation of existing equipment.

Conflict and confrontation in the PA infrastructure was not uncommon either between, or within, different PA groups. In this study, several participants reported that they had encountered both overcrowding of the PA spaces and/or conflict with other users when taking part in PA. The pandemic led to an increased interest in and engagement by people in PA, and the COVID-19 restrictions meant that there was a concentration of greater numbers of people into fewer and smaller available spaces ([41]). This may have increased the likelihood that conflict would occur within and between different PA groups. Participants reported that lack of etiquette and adherence of protocol when undertaking PA such as sharing the space; people refusing to keep to one side of the paths; people exercising side-by-side, not social distancing and failing to keep their dogs under control on shared paths; exercisers wearing noise cancelling headphones that could not hear other users approaching; looking at mobile phone and/or not paying attention to what was going on in the surroundings were not uncommon. The outcomes sometimes included disagreements, arguments and occasionally verbal abuse from other users. Further, participants reported feeling that some novice exercisers did not know and or demonstrate established or accepted protocols when physically active, such as cycling on the left (in the UK), or giving way and letting faster runners, wheelers or walkers pass or just not paying attention to what was going on in the PA environment. In some cases, there were reports of pedestrians telling cyclists and other wheelers to ‘get off’ permissive multi-mode facilities. In these instances, the pedestrians were using combined walking and cycle routes for the first time and did not know it was a ‘shared’ facility. Participating in PA is not only about the core behaviour of PA, but also about learning and refining the behaviours that support PA, including the etiquette and protocol, and this can take time to learn and refine for novice exercisers. While some protocols are promoted through signage and messaging, in other cases, the rules and protocols are unwritten ([47]) and become established through regular participation and engagement with others in such spaces.

Several Actives were pleased to see so many new recruits adopting PA and experiencing the benefits of a physically active lifestyle ([46]). However, in this study, there was also a feeling by some Actives that some novice exercisers had only taken up PA because of the unavailability of their usual sedentary pursuits and some reluctantly accepted, or did not always welcome their presence. Some felt that their presence in the PA space was transient, and that once COVID-19 restrictions eased and hospitality, retail and travel opened, new exercisers would soon discard their newly adopted PA and return to their more sedentary leisure time pursuits. This example reflects the tensions that exist with and between PA groups, which are rarely considered during the promotion of PA. Research has shown that purchasing expensive exercise kit, such as cycling equipment, supports a desired identity, appearance and lifestyle ([8]). It should be remembered that not everyone will be inclusive of newcomers to the PA scene. Social support, compassion and a harmonious social environment are important for a welcoming and inclusive PA space ([35]). People often need help in the PA space, such as directions to and from places, route knowledge and advice or help with a bike repair or medical help, yet some of the tensions reported here may detract from this communal aspiration going forward. Further, several activities, PA events and challenges are built on the premise of inclusivity and community, such as mass participation events, challenges and taster activities, so this is an important consideration.

Participants also reported that local places they used for PA were occasionally made to feel unwelcoming by residents who lived in and near them, including residents who told Actives that they could not use these spaces because they were not ‘a local’. In some instances, residents generated unofficial posters and signs instructing people not to use the places and to ‘go home’. In more extreme instances, our participants told us of reports of people making signs look as if they were from the police force to make the messaging look more authoritarian and official, as well as local people putting glass and nails on the road to stop cyclists using local roads. Participants also told us that some local people were actively monitoring people’s use of these PA spaces and threatening to inform the authorities, such as the police. There was the feeling that some of these people were taking pleasure from surveying and trying to control space, even though they had no jurisdiction to act in such a manner. Several participants commented on how irrational this behaviour was and explained the behaviours of these individuals as symptomatic of COVID-19 times. These types of events can foster resentment, remain in the memory of participants and can rail against aspirations for a hospitable, caring and compassionate PA environment in the future.

These interactions were stressful for participants at what was an unprecedented time of difficulty and when people reported negative impacts on their mental health ([35]). Several participants told us how they subsequently avoided areas that were being unofficially ‘policed’ or where they had previously experienced conflict and kept a low profile to not attract attention to their presence and even exercised during times in the early hours or at night, as seen elsewhere. Further, some Actives reported they had been subject to micro-aggressions, including verbal abuse. Microaggressions have been defined roughly ‘as communicative acts denigrating an individual by targeting perceived social aspects of their identity’ ([16]). Exceptionally, some women in this study reported where they had been subject to abuse and intimidation by other users, such as groups of men refusing to share the space or making derogatory remarks based on gender. Research has identified that women experienced a perceived threat of harassment, intimidation and/or violence when using the PA space ([24]). Research identified that in women cyclists who have been subject to micro-aggressions, over half of the participants reported feelings of anger or frustration due to the microaggressions, followed by feelings of sadness ([4]). Helping people to negotiate and manage these events is an important part of helping people start and keep active. A notable absentee in the findings was injury, which is a feature of being active. This only occasionally presented in the data, which could reflect that this was not a significant barrier to PA.

### 4.3. Facilitators to Physical Activity

[12] ([12]) identified that parks and green spaces contribute to a healthy society. Several of our participants had access to green space in urban and rural locations and were able to utilise this to help facilitate their PA. Indeed, the convenience of spaces to be active has been identified as a route through to PA participation ([35]) for active and inactive older adults ([44]). Participants reported access to spaces and places for PA for both longer and shorter bouts of PA, including ‘snack activity’ and small bouts of PA such as ‘walking around the block or walking around a back garden’. This was especially important for breaking up sedentary time when people were working from home. Indeed, the UK CMO guidelines highlight the importance of breaking up prolonged periods of sedentary behaviour with light activity such as that reported in this study; these are important for health and wellbeing ([46]), so convenient routes and pathways to do this were important for some of our Actives. However, it is important to recognise that many people who lived in city flats perceived the lack of space for PA as a substantial barrier that negatively impacted their PA and mental health ([12]), and that outdoor PA is an important part of people’s wellbeing ([44]) including mental health ([46]). In contrast, our participants reported that their rural geographical location and living on the edge of open moorland sometimes helped facilitate PA in locations where it was rare to encounter another person. In other examples, Actives reported searching for new locations that were not commonly known or populated by other people, sometimes covering huge distances to locate isolated places.

Associated with outdoor PA participation during the first months of the pandemic was the weather, Actives reported the clement weather that coincided with the first government lockdown as a facilitator for PA. Indeed, warm temperatures for the months of March-May 2020 and increased hours of daylight in the springtime helped to facilitate PA and this was reported by our participants.

All our Actives reported the processes they adopted to keep physically active during the pandemic. When we consider the importance of sedentary and inactive populations learning and refining such behaviours to help cement PA into daily habitual routines, one such facilitator was establishing a routine, where PA was scheduled into a daily regimen around other commitments. Research has shown that ‘while inactive participants had been busy with home-based activities such as cooking and cleaning, the lockdown restrictions awoken a need to be more purposeful in their physical and mental pursuits’ ([44]). Participants reported structuring their day around PA, such as going for a run, performing some interval training or some circuit training in some instances, like a break from work or childcare.

Although a loss of social support was a barrier for some participants, changes to people’s living arrangements led to new opportunities to support PA. Our Actives reported how older children had moved back in with them at the family home and they undertook PA together, or how they established new social connections with local people to eventually undertake PA when restrictions eased. In some instances, social connections became a form of accountability and contract to perform PA together.

Several of our Actives reported substituting their usual PA with a replacement, including using online platforms for exercise, where their local instructor, delivering virtual classes online, using a pre-recorded video on the YouTube channel, such as the Global Cycle Network sessions to replace commuting by cycling and interactive real time classes in virtual environments. Research has shown that digital platform users were more likely than non-users to meet PA guidelines during the COVID-19 stay-at-home restrictions in April and May 2020 ([30]). Indeed, such provision has been shown to facilitate PA participation in Actives with chronic health conditions, such as cancer ([35]). Actives also reported making changes to their local environments, or family members making changes, such as making a walking/running track in their garden, using their front room for exercise classes and putting a turbo trainer on their patio. This was not always to everyone’s preferences, with some Actives unhappy about their living room being used as a gym by other family members, but it reflects the improvisation Actives demonstrated to keep active.

Goal setting, setting challenges and monitoring, were also strategies reported by participants, such as setting the number of miles run, a suite of PA challenges throughout the year with family members, either in person or remotely, and keeping a record of distances covered in a set time as a record to show progress in fitness, while all helped to keep PA interesting and varied for our Actives. In some cases, it became a ‘raison d’être’ to demonstrate to others that they had been proactive in maintaining their exercise participation.

### 4.4. Government Guidelines and Messaging for Physical Activity and Health

The parameters of what PA was permitted under the Government COVID-19 guidelines were not always seen as helpful by our participants. This included a lack of clarity of what was meant by ‘staying local’ and the duration of PA that was permitted under the exercise ‘once per day’ rule. Guidelines are an important channel for the promotion of PA and health ([36]), and research has identified that inactive people have reported not being aware of the PA recommendations, but also appreciate guidance on the frequency, intensity, type and time of PA ([44]). Regarding the promotion of PA, some participants also felt that the opportunity was lost to promote the benefits of PA to all groups through the messaging that took place during the pandemic. The UK CMO PA guidelines ([46]) highlight the benefits of strength and balance for adults undertaken at least twice a week, yet some participants felt more could have been undertaken in this respect, especially on messaging the importance of PA, strength and balance to older people. Studies on PA during the pandemic confirm its positive impact on mental health. The impact of COVID-19 had negative consequences for mental health, with reports of increases in anxiety, depression and loneliness ([11]; [22]). As reported in this study, some participants said that their mental health had impacted their motivation to undertake PA. Research has also shown this, where people have felt ‘a bit down’, did not feel like it or ‘anxious’ to do PA ([44]). Regular PA during the pandemic relieved psychological stress, improved people’s mental state and effectively enhanced the immune system ([27]). Given the impact of the pandemic on mental health, participants felt that the guidance and messaging at the time could have been better promoted, including the benefits of PA for positive mental health ([46]) such as fun, enjoyment and feelings of wellbeing ([44]). While others reported the need for guidance on where people could find information on how to be physically active.

[21] ([21]) has argued that the achievement of the PA guidelines will not happen because there is a PA guideline, but if the guideline is genuinely invested in, well-resourced and has political support. During the pandemic, significant investment was made in the communication of the COVID-19 guidelines, and this was also an opportunity to promote PA, and the different benefits associated with being active ([46]). Indeed, COVID-19 guidelines were promoted across multiple platforms, media and broadcasters. It has been argued that the promotion of PA guidelines requires a comprehensive communication strategy which would include expert social marketing and a diverse range of media platforms, including TV, radio, social media, on-demand and web-based media ([28]). While some promotion of PA took place through some of these channels, for example on Sky Sports News, this was an opportunity to really promote the specific benefits of PA and to different groups. Moreover, for those people who adopted and or maintained a PA routine during the pandemic, there was an opportunity to encourage people to cement the habitual nature of an active lifestyle and processes to support PA ([23]) in their everyday lives, following COVID-19 and the easing and cessation of restrictions.

Participant adherence to the COVID-19 guidelines varied. Some of our participants reported strictly following the messaged directives of exercising once per day, whereas other participants had a more liberal interpretation of the guidance. For example, participants who lived in rural and remote locations were afforded much greater flexibility, because their local area was not only sparsely populated, but also because they lived in rural and remote areas that people would normally travel to. As such, the chances of participants meeting other people were unlikely and the risk of transmission of the virus was low. Moreover, these locations were less likely to be policed. In some instances, participants reported that their PA participation was ‘like before’ the COVID-19 pandemic commenced, while others who adhered to the guidelines felt a sense of injustice that other people were ‘flouting the rules’. Indeed, we interviewed participants who ‘went out to work’ in the NHS and other essential services during the pandemic, when the risks of contracting the virus were very high. Participants reported being unhappy that some people were not following the COVID-19 guidelines ([13]). This was exacerbated further when those participants were able to take part in PA in their free time and days off and found that the PA infrastructure was overcrowded and people were not adhering to the directives, such as social distancing or the etiquette of being active such as exercising in large groups and or not sharing the PA space.

There is much focus in contemporary public health on viewing challenges and solutions to PA promotion using a social–ecological perspective ([1]; [29]). The social–ecological model (SEM) ([17]) is a useful lens for understanding and intervening on the wider determinants impacting on health and PA and is a feature of current approaches for PA promotion. In the UK and as part of Uniting the Movement, Sport England’s 10-year PA strategy ([43]), an investment has been made in place-based approaches to PA promotion ([29]). The approach aims to support PA at different levels using a socio-ecological perspective, from the individual to the social environment, including communities; organizations; institutions; the physical environment, including blue, grey and blue spaces; and the policy environment including programmes, promotion, guidelines and legislation ([17]). In considering the findings from this study, while the SEM was not explicitly used to organise this research per se, the different layers are useful when interpreting the findings more broadly. When considering the barriers and facilitators that our Actives experienced, these are multi-layered, from the individual through to the social, physical and policy environmental levels. What is especially interesting is the impact that the social environment had on participants’ experiences and PA, especially the engagement of people and their local environs. This includes social support and changing social dynamics, as well as conflict, confrontation and overcrowding in the PA space. Linked to this, the physical environment was an influential enabler or inhibitor, including where people lived or worked. The (un)availability of services and spaces for PA also acted to facilitate and inhibit participation. The policy environment was similarly important, including the government restrictions, what people could and could not do, the guidance and messaging for PA and people’s interaction with this at an individual level. As such, it is important to consider the different layers of determinants when considering the practical implications for PA policy and promotion.

### 4.5. Strengths and Limitations

This research has several limitations. These include its small convenience sample, which limits generalization. Participants were white British, professional and predominantly an adult/older adult sample. Reports would have been different for a more diverse representation of participants from different groups and locations. In the interview schedule, we asked participants about their PA in different lockdown phases, and on occasions, it was difficult for some people to distinguish between these. As such, we reported their experiences across the period of government restrictions. No objective data were collected on how many minutes per week participants were active prior to and during the lockdowns. This, along with more comprehensive demographic profiles of participants, is a consideration for future survey research. However, we do provide examples of qualitative accounts from participants on the frequency, intensity, time and type of PA they participated in, as planned, in this study. Study strengths include the depth, candidness and insightfulness of the accounts of the experiences of Actives during the COVID-19 pandemic. These include honest reports of conflict and confrontation that participants experienced, how people felt about what they witnessed in the PA spaces and adherence to the COVID-19 guidelines, as well as the determinants that they faced in performing PA. Importantly, the study provides insights into the strategies that active participants deployed to keep active during this period that can be helpful in shaping strategies to help people keep active. Further, the study adapted an interview schedule that has been subject to an independent peer review ([35]) and an internationally recognised protocol for thematic analysis of the data ([7]). The outcomes from the study provide several practical recommendations for PA promotion that will be shared through regional PA networks, such as Making Our Move ([1]) and the National Evaluation Learning Partnership, (2025). Moreover, the interviews will help shape questions for a larger quantitative survey into this topic in a more informed way, which would not have been possible had this study not been undertaken.

## 5. Conclusions

Actives encountered barriers to PA participation, meaning that, in some instances, they were required to reduce, adapt or substitute their PA participation. In other cases, participants continued with the same mode of PA and even increased their PA levels. Overcrowding and conflict with other users were among the most significant barriers to participation. At times, it was difficult for some of our participants to keep physically active, and it is important to recognise that Actives also experience challenges in maintaining PA and need support to help them to maintain their PA. Actives adopted strategies to overcome barriers and to keep active. Participants also reported how messaging and guidelines could have been improved to help promote the benefits of PA participation, as well as what was permissive during this period. Our study provides valuable implications for practice.

## 6. Practical Implications

It is important that research informs intervention design and delivery ([34]). Reflecting the social–ecological perspectives discussed earlier ([17]), this research has several practical implications for PA promotion, policy and intervention.


Messaging of PA guidelines should promote the benefits of activities that promote strength and balance, especially to adults and older adults, as well as the benefits of PA for mental wellbeing.Messaging should also direct people to resources on how to be physically active and the protocols and processes for being active in PA spaces.Promotion and messaging of small doses of PA, which are important for breaking up sedentary time, health and promoting feelings of wellbeing ([25]).Promotion of the behaviours that support PA, including goal setting, monitoring progress, scheduling a routine, planning and self-management.Conflict and confrontation feature in the PA space; providing some guidance on how to avoid and manage these types of events can be helpful when engaging in the PA infrastructure for enjoying the benefits of PA.Understanding the complexities associated with conflict and confrontation can help with making PA spaces more attractive for PA.Compassion and consideration of others are important for a more harmonious PA space, including guidance on the considerate use of these spaces, such as sharing infrastructure and observing the UK Highway Code [or the international equivalent] ([14]). This is especially important given the multi-million-pound investment in Active Travel ([2]). It is important to not only consider the physical infrastructure, but also how people interact within the environment.


## Figures and Tables

**Table 1 behavsci-15-00598-t001:** An overview of themes and sub-themes.

Themes				
Physical Activity	Barriers to PA Participation	Facilitators for PA Participation	Strategies Adopted for PA Participation	Government Guidelines and Messaging
Sub-Themes				
PA Levels	Closure of spaces and places for PA	Making the most of opportunities to be active	Engaging in physical activity challenges	The guidance could be more helpful for promoting PA
Decreases in PA	Lack of availability of products to support PA	Place of residence	Setting goals for physical activity participation	A lack of guidance on strength and balance training
PA Substitutions	An overcrowded PA space	Additional permissions to leave home for PA	Keeping physical activity interesting	A lack of guidance on mental health
Increases in PA	Novice exercisers and not sharing space	Mental health	Online physical activity provision	A lack of clarity on what was defined as ‘local’ exercise
	An unwelcoming space for PA	Weather	Skills and processes to support PA participation	Guidance on how often and long people were allowed to exercise was interpreted differently
	Loss of social support for PA	Social support	Keeping a low profile to avoid conflict with local people	

## Data Availability

The datasets presented in this article are not readily available due to privacy restrictions.

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
