# Peer review of "An Investigation into the Perspectives and Experiences of Physically Active Adults During the COVID-19 Pandemic"

_behavsci, 2025, doi:10.3390/bs15050598_

Round 1

Reviewer 1 Report

Comments and Suggestions for Authors

Manuscript ID: behavsci-3536637

Title: An Investigation into the Barriers, Facilitators and Strategies Active 
People Experienced/Deployed to Keep Active during the COVID-19 Pandemic

The manuscript presents a qualitative study examining barriers and facilitators of engaging in PA during the COVID-19 pandemic in a sample of active adults. Thirteen active participants completed semi-structured interviews discussing facilitators and barriers to engaging in PA during the COVID-19 pandemic. Five themes were identified discussing changes in physical activity, facilitators, and barriers to engaging in physical activity during the pandemic as well as strategies supporting PA, and government messaging. The manuscript was clear, easy to read, and provides insights on physical activity behavior during a unique period, however, more clarity on the study purpose and alignment with existing behavioural theories is needed. I hope the following comments may help to refine the manuscript:

General comments:

  • A clear purpose statement would be beneficial as there is sometimes misalignment between the study rationale/purpose, research question, and results. Specifically, it was unclear whether the purpose was to identify 1) barriers/facilitators of PA in active groups to support active individuals, 2) barriers/facilitators of PA in active groups to support inactive individuals, 3) strategies for keeping active during COVID-19, or 4) to describe the behavioural changes in people who were active pre-pandemic. The instrumentation section mentions data were collected on barriers and facilitators to PA participation and strategies for keeping active (L 189-191), however it was also mentioned that “…learning from the experiences of physically active people is also helpful in sharing the strategies they deploy to maintain PA and can be helpful for supporting sedentary and inactive groups to sustain their PA”(L 159-160). Furthermore, the themes of ‘Physical activity levels’ and ‘Government Guidelines and Messaging’ do not seem to align with barriers, facilitators, or strategies but rather experiences with engaging in physical activity during the pandemic more broadly. Better alignment with study purpose, methods, and results would help improve the overall clarity of the manuscript.

  • I think the manuscript would be strengthened with some discussion of how the research question(s) and study results are informed by/align with existing behavioural theories. Some potential theories include the Socioecological model (doi: 10.1177/109019818801500401) given the impact of policy on PA during the pandemic, the Multi-Process Action Control Framework (doi: 10.3389/fpsyg.2021.797484) which discusses reflexive influences such as habit and identity on PA, or the Health Action Process Approach (doi: 10.1027/1016-9040.13.2.141) which focuses on post-intentional factors influencing behavior.

  • The authors argue throughout that understanding the needs of physically active groups can help sedentary and inactive people to adopt and maintain physical activity. I think further justification for this premise is needed and I would argue the motivational profiles of inactive vs. insufficiently active vs. active groups are different with unique needs. For example, inactive people may lack the knowledge or motivation for the behaviour and benefit most from interventions targeting knowledge or reflective motivation to strengthen intentions. Insufficiently active individuals may possess the knowledge and intention but lack the behavioural regulation skills to regularly engage in physical activity. Active individuals may benefit most from strategies targeting reflexive motivation. Further justification is needed.

Specific comments:

  • The introduction describes rates of inactivity among older adults (aged 55-84; L37-39) which does not include the age range in the present study (18-74.) Please update the introduction with reference to inactivity rates in younger adult populations. Also, consider including a mean age of study participants for descriptive purposes.

  • Is there any additional demographic information to describe the sample? All were “active” but is there data on ~how many minutes/week they were active prior to and during the lockdowns? Details on the activities they participated in prior to the lockdowns would also be helpful to determine generalizability. For example, cycling seems to be a common theme from participant quotes which makes me wonder if findings from this study are more applicable to individual aerobic physical activities vs. group or team based sports (e.g., soccer) or resistance physical activity.

  • Can the authors provide details on the average duration of the interviews? Also, were participants provided remuneration for their participation in the study?

Author Response

Thank you for your suggestions.   We have detailed our response below and these are highlighted in the manuscript in yellow.

Comment 1: 

A clear purpose statement would be beneficial as there is sometimes misalignment between the study rationale/purpose, research question, and results. Specifically, it was unclear whether the purpose was to identify 1) barriers/facilitators of PA in active groups to support active individuals, 2) barriers/facilitators of PA in active groups to support inactive individuals, 3) strategies for keeping active during COVID-19, or 4) to describe the behavioural changes in people who were active pre-pandemic.

Response:

Page 4/Line 171- We have revisited the introduction to clarify the purpose of the research.  We have now said This study investigated perspectives and experiences of physically active adults (‘Actives’) during the COVID-19 pandemic, including the barriers and facilitators to/for PA, the strategies they deployed to keep active and their experiences of the messaging of Government health and PA guidelines.  

Page 1.   We have reviewed and updated the title of the paper to make this more expansive and inclusive of all the areas included in this study.  The title now reads: An Investigation into the Perspectives and Experiences of Physically Active Adults During the COVID-19 Pandemic

Page 1/Line 11-15 we have updated the abstract to reflect the revised title and how this aligns with the areas investigated in this study. We have added: This study investigated perspectives and experiences of physically active adults (Actives’) during the COVID-19 pandemic, including their PA levels, barriers and facilitators to/for PA, the strategies they deployed to keep active and their experiences of the messaging of Government health and PA guidelines

Comment 2: 

Furthermore, the themes of ‘Physical activity levels’ and ‘Government Guidelines and Messaging’ do not seem to align with barriers, facilitators, or strategies but rather experiences with engaging in physical activity during the pandemic more broadly. Better alignment with study purpose, methods, and results would help improve the overall clarity of the manuscript.

Response: 

Page 4/Line 171- We have revisited the introduction to clarify the purpose of the research. This study investigated perspectives and experiences of physically active adults (‘Actives’) during the COVID-19 pandemic, including the barriers and facilitators to/for PA, the strategies they deployed to keep active and their experiences of the messaging of Government health and PA guidelines.   

Page 1.   We have reviewed and updated the title of the paper to make this more expansive and inclusive of all the areas included in this study.  The title now reads: An Investigation into the Perspectives and Experiences of Physically Active Adults During the COVID-19 Pandemic

Page 1/Line 11-15 we have updated the abstract to reflect the revised title and how this aligns with the areas investigated in this study. We have added: This study investigated perspectives and experiences of physically active adults (Actives’) during the COVID-19 pandemic, including their PA levels, barriers and facilitators to/for PA, the strategies they deployed to keep active and their experiences of the messaging of Government health and PA guidelines

In the Appendix 1/Page 32, We have also added the interview schedule to provide the reader with a better understanding of the questions that were deployed under these categories.   

Comment 3: I think the manuscript would be strengthened with some discussion of how the research question(s) and study results are informed by/align with existing behavioural theories. Some potential theories include the Socioecological model (doi: 10.1177/109019818801500401) given the impact of policy on PA during the pandemic

Response: 

Page 21/Line 904-30.   Thank you for this suggestion, in the discussion, we have added some additional content on the SEM to bring the discussion together. While we did not use SEM to organise the research, it is a useful framework for drawing together the findings, notably the barriers and facilitators and strategies and we highlight the importance of this when considering the practical implications.   Our practical implications section also make reference to the SEM.

We have said: There is much focus in contemporary public health on viewing challenges and solutions to PA promotion using a Social-Ecological perspective (Active Derbyshire/Active Notts, 2022; National Evaluation Learning Partner, 2025). The Social-Ecological Model, (SEM), (Eldredge et al., 2016) is a useful lens for understanding and intervening on the wider determinants impacting on health and PA, and is a feature of current approaches for PA promotion. In the UK and as part of Uniting the Movement, Sport England’s 10-year PA strategy (Sport England, 2022), an investment has been made in place-based approaches to PA promotion (National Evaluation Learning Partner, 2025). The approach aims to support PA at different levels using a Socio-Ecological perspective, from individual to the social environment, including communities, organizations, institutions, the physical environment, including blue, grey and blue spaces and the policy environment including programmes, promotion, guidelines and legislation (Eldredge et al., 2016). In considering the findings from this study, while the SEM was not explicitly used to organise this research per-se, the different layers are useful when interpreting the findings more broadly. When considering the barriers and facilitators that our ‘Actives’ experienced, these are multi-layered, from the individual through to the social, physical and policy environment levels. What is especially interesting is the impact that the social environment had on participants’ experiences and PA, including social support and changing social dynamics, as well as conflict, confrontation and overcrowding in the PA space. Linked to this, the physical environment was an influential enabler or inhibitor, including where people lived, or worked, the (un)availability of services and spaces for PA also acted to facilitate and inhibit participation. The policy environment was similarly important, including the government restrictions, what people could and could not do, the guidance and messaging for PA and people’s interaction with this at an individual level. As such, it is important to consider the different layers of determinants when considering the practical implications for PA policy and promotion

Page 23: Line 978 Our practical implications section also make reference to the SEMWe said: It is important that research informs intervention design and delivery (Pringle et al., 2020). Reflecting the Social Ecological perspectives discussed earlier (Eldredge et al., 2016), this research has several practical implications for PA promotion, policy and intervention.

Comment 4: The authors argue throughout that understanding the needs of physically active groups can help sedentary and inactive people to adopt and maintain physical activity. I think further justification for this premise is needed and I would argue the motivational profiles of inactive vs. insufficiently active vs. active groups are different with unique needs. For example, inactive people may lack the knowledge or motivation for the behaviour and benefit most from interventions targeting knowledge or reflective motivation to strengthen intentions. Insufficiently active individuals may possess the knowledge and intention but lack the behavioural regulation skills to regularly engage in physical activity. Active individuals may benefit most from strategies targeting reflexive motivation. Further justification is needed.

Response: Thank you, we have now removed the focus on sedentary and inactive groups and kept the focus on active adults.

Reviewer 2 Report

Comments and Suggestions for Authors

Include in the materials and methods section that a qualitative research method was used, along with appropriate justification.

Include the interview script in the appendices to provide a clear and detailed overview of the process. This will help assess the consistency between the study’s objective and the information obtained, facilitate the replicability of the study, acknowledge the contextualization of the results, and justify the analysis performed. Ultimately, this will strengthen the validity, reliability, and transparency of the research, making the results more comprehensible and useful to the scientific community.

Clearly outline the category system (codes) used, demonstrating how the analysis was structured and the criteria applied to interpret the data. This will help determine whether the conclusions drawn are well-founded.

Explicitly state in the text that an inductive categorization was carried out, as can be inferred from lines 196 and 197.

Include a clear definition of the meaning of the codes and themes used to facilitate their interpretation by readers.

Author Response

Thank you for your helpful suggestions. Our responses to your comments are included below.  These are highlighted in yellow in the manuscript.

Comment 1: Include in the materials and methods section that a qualitative research method was used, along with appropriate justification.

Response: Page 5/Line 190-195. We have added that a qualitative approach was used and this has been used to effectively collect information on the PA behaviors and barriers and facilitators of Actives during the COVID-19 pandemic previously (Pringle et al., 2022; Rutherford et al., 2021). A copy of the interview schedule used is available in Appendix 1.

Comment 2: Include the interview script in the appendices to provide a clear and detailed overview of the process. This will help assess the consistency between the study’s objective and the information obtained, facilitate the replicability of the study, acknowledge the contextualization of the results, and justify the analysis performed. Ultimately, this will strengthen the validity, reliability, and transparency of the research, making the results more comprehensible and useful to the scientific community.

Response: Page 30    The Interview schedule has been added please see appendix 1.

Comment 3:  Clearly outline the category system (codes) used, demonstrating how the analysis was structured and the criteria applied to interpret the data. This will help determine whether the conclusions drawn are well-founded.

Response: Page 5/Line 219  In the results, we have included Table1 to show the themes and alignment with the sub-themes with the aim of providing an overview of the results.   

Comment 4: Explicitly state in the text that an inductive categorization was carried out, as can be inferred from lines 196 and 197. 

Response: Page 5/Line 218 Thank you, we have added that inductive categorization was carried out. 

Comment 5: Include a clear definition of the meaning of the codes and themes used to facilitate their interpretation by readers.

Response: Page 5/219 Thank you for this suggestion, we have added a table 1 that shows the themes and sub-themes relationship between the themes and sub-themes used to organise the interview excerpts.   

Comment 6:  Include in the materials and methods section that a qualitative research method was used, along with appropriate justification.

Response Page 5/Line 190-195  We have added  This approach has been used to effectively collect information on the PA behaviors and barriers and facilitators of Actives during the COVID-19 pandemic previously (Pringle et al., 2022; Rutherford et al., 2021).    Page 5 197: A qualitative interview approach was adopted in this study, and this has been used to obtain rich informative accounts of the PA characteristics of participants, including their experiences, barriers and facilitators to PA

Reviewer 3 Report

Comments and Suggestions for Authors

I congratulate the authors on their pertinent topic. Its structure and organisation is clear and well defined. The methodology is clear and well-founded. The results and conclusions are coherent.  I suggest considering a more structured presentation of the results.

Author Response

Comment 1: I congratulate the authors on their pertinent topic. Its structure and organisation is clear and well defined. The methodology is clear and well-founded. The results and conclusions are coherent.  I suggest considering a more structured presentation of the results.

Response:

Line 216 Page 5: We have added Table 1 which provides and overview of the Themes and Sub-Themes and will help the reader navigate the findings .  Thank you for your positive comments on the manuscript.